# Cytotoxicity Assessment of PM_2.5_ Collected from Specific Anthropogenic Activities in Taiwan

**DOI:** 10.3390/ijerph16245043

**Published:** 2019-12-11

**Authors:** Tuan Hung Ngo, Pei Chun Tsai, Yune-Fang Ueng, Kai Hsien Chi

**Affiliations:** 1Institute of Environmental and Occupational Health Sciences, National Yang Ming University, Taipei 112, Taiwan; tuanhung0712@gmail.com (T.H.N.); poipeiun@gmail.com (P.C.T.); 2International Health Program, National Yang Ming University, Taipei 112, Taiwan; 3Divisions of Basic Chinese Medicine, National Research Institute of Chinese Medicine, Taipei 112, Taiwan; ueng@nricm.edu.tw; 4Institute of Medical Sciences, Taipei Medical University, Taipei 112, Taiwan

**Keywords:** PM_2.5_, cell toxicity, long-range transport, traffic activity, cooking activity

## Abstract

Fine particulate matter (PM_2.5_) from different sources with different components have different health impact. In this research in Taiwan, composition and cytotoxicity of PM_2.5_ from long-range transport event (LRT), traffic activity, and outdoor cooking at night market were studied. The PM_2.5_ mass concentrations were 39.0 μg/m^3^ during LRT, 42.9 μg/m^3^ at traffic area, and 28.3 μg/m^3^ at the night market. Traffic area had highest concentrations of PCDD/Fs (46.9 fg I-TEQ/m^3^) when highest PAH concentrations of 3.57 BaPeq-ng/m^3^ were found at night market area. One quarter of PM_2.5_ mass at LRT and night market was constituted by water-soluble ion (26.02–28.93%). Road dust (represented by high concentration of Al and Ca) was the main contributor for metal element at traffic station whereas presence of natural salt (Na and Cl elements) was a marker of LRT and cooking activities. Cell viability reduced 9% after exposure to organic extracts of 0.316 μg of PM_2.5_ from LRT and night market samples. 150% elevation of ROS production was observed after exposure with organic compound of night market samples at the dose equivalent to 10.0 μg PM_2.5_. Organic extracts from night market induced positive genotoxicity in *umu* test (at a dose of 20.0 μg PM_2.5_).

## 1. Introduction

Fine particulate matter (PM_2.5_) is the most ubiquitous pollutant that can heavily influence the health of humans. The effect of particulate matter on health can be altered by their size. The smaller size of the particles, the more severe health impact they can cause. PM_2.5_ is considered as respirable particles as they can reach until the gas-exchange region of the lung [1]. Once PM_2.5_ penetrates into the deepest part of the respiratory system, many potential health hazards can be caused. PM_2.5_ intoxication can first harm respiratory system leading to reduced lung function or even lung cancer [2,3]. Increment of ambient air PM_2.5_ concentration was also found to increase the risk factor of cardiovascular diseases [4,5]. In contrast, reduction of ambient air PM_2.5_ concentration can improve the lung function [6] and reduce mortality [7].

Particles themselves also carry many different types of pollutants. Small size particles such as PM_2.5_ can possess much higher surface area than that of bigger particles of the same mass. Metal elements and ions are known to cause stress to the cell which in turn leads to inflammation [8]. Other organic compounds such as polycyclic aromatic hydrocarbons (PAHs) and polychlorinated dibenzo-p-dioxins and furans (PCDD/Fs) were found to be carcinogenic in previous studies [9,10].

The compositions of pollutants bounded on the surface of PM_2.5_ are influenced by their emission sources. Looking at metal proportion, higher concentrations of Fe, Ca, Na, Mg, Al, and K are considered to relate to soil dust [11,12]. The elevation of Ca suggests the presence of construction site in the vicinity of the sampling stations. Concentrations of Cu, Zn, Sb, Pb, and Ni are indicators of vehicle exhaust pollutants [11]. On the other hand, different thermal activities of stationary sources contribute differently to the components of organic compounds such as PAHs or PCDD/Fs in PM_2.5_ [13].

In the guideline of World Health Organization (WHO), the standard ambient concentration of PM_2.5_ was well defined to be 10 µg/m^3^ annually or 25 μg/m^3^ daily [14]. However, there has not been any guideline about the component of pollutants attached on PM_2.5_. Studies about characteristics of PM_2.5_ in ambient air have found the different concentrations of chemical compounds in different sources of PM_2.5_. Metals and ions were important elements in research of characteristics done in developing countries in Asia [15,16,17]. However, organic compounds such as PAHs and PCDD/Fs are gaining attentions. This suggested the necessity to carry out research on others factors aside from PM_2.5_ mass itself influencing health outcome.

Epidemiological studies suggest that long-term exposure to air pollution can increase the risk of cardiovascular diseases and respiratory diseases [18,19]. The concentration of PM_2.5_ in Taiwan highly varied based on location. A measurement in the north of Taiwan found mean PM_2.5_ concentrations of 17.7 ± 13 μg/m^3^ [20], whereas those concentrations were found to be 29.3–102.2 μg/m^3^ in central Taiwan [21] and 43.2 ± 20 μg/m^3^ in southern Taiwan [22]. The atmospheric PM_2.5_ of Taiwan was significantly influenced by three main sources including long-range transport, traffic, and industrial activities. On the other hand, night market is a daily activity in Taiwan attracting participation of large number of people. The night market is the place where local people often go to look for food. Therefore, cooking activities at these locations are usually intense and concentrated, especially during meal hour. Since industrial activities are normally located distance away from people residency, the effect of this source of pollution on human health should be specially studied. In this research, we carried out the study focusing on the toxicity of PM_2.5_ from long-range transport event, traffic, and cooking at the night market sources.

Many studies were carried out in different biological models to determine the toxicity of PM_2.5_, among those, cell model was one of the most effective models since it is fast and easy to show the outcome. Studies found that after exposure to PM_2.5_, the development of cell declines. Therefore, we aim to elucidate the effect of PM_2.5_ (including metal, ion, and organic compounds bound on it) originating from LRT event, traffic activities, and open cooking activities in the night markets on cell development.

## 2. Materials and Methods

### 2.1. Sampling Location and Sampling Methods

PM_2.5_ samples were collected in 2016 from three stations representing three major sources of pollutants in Taiwan (Figure 1).

Fugui Cape is a land cape located in the North peak of Taiwan. Because of its special location, Fugui Cape is the first place of Taiwan exposed to long-range transport pollutant when winter monsoon brings pollutants from the northern area to Taiwan [20]. The samples at this area were collected during the day when significant monsoon activity was recorded. Zhongshan station was located in the downtown of Taipei with high loading of traffic and serve as representative of urban area in Taiwan. Taiwan has one of the highest motorcycle densities in the world at the rate of 60 motorbikes per 100 people. In our research area of Taipei and New Taipei area, approximately 5 million personal transportation (cars and motorbikes) were registered in 2018 [23]. PM_2.5_ samples at Zhongshan station was collected on ordinary weekdays. On the other hand, in Taiwan, night market is a part of the culture of the island where food can be freshly prepared on site. The common food in Taiwan’s night market are barbecue and deep-fried, which produce and disperse high amounts of fine aerosols during cooking. Every night, thousands of people, both local and foreign tourists, spend hours at the night market getting exposed to cooking fumes. In this research, we collected PM_2.5_ samples at a night market in New Taipei city during peak hours in the weekend. The term night market samples and cooking samples are used alternatively in this article.

In accordance with the European Committee for Standardization standard for PM_2.5_ (EN 14907), the sampling of PM_2.5_ was conducted at a flow rate of 500 L min^−1^. All PM_2.5_ samples collected at LRT and traffic stations were obtained from 24-hour sampling. Because of the short opening time, the night market samples were only collected for 6 h. The high-volume samplers for PM_2.5_ (Analitica AMS^®^ Air Monitoring System, PM_2.5_-HVS) captured particles on quartz fiber filters (ADVANTEC^®^, http://www.advantec.tw/, QR-100, ⌀150 mm). Before sampling, the quartz fiber filters were baked at 900 °C for 5 h, conditioned at a constant humidity of 45% ± 5% and a temperature variation of <3 °C for at least 24 h, and weighed on a scale with a sensitivity of 0.1 mg. We collected two sample filters at each station.

### 2.2. Chemical Analysis

After sampling, all filter samples were conditioned in standard condition in the laboratory (20 °C, relative humidity = 40%). The mass of PM_2.5_ was determined using gravimetric analysis. One-sixteenth of each filter underwent ion component analysis, another one-sixteenth underwent metal analysis, when six-sixteenths was used for organic chemical analysis. The remaining filters were extracted and used for cell exposure study and backup (Appendix A). The on-site blank filters (unsampled filters) were served as control samplers for QA/QC and cytotoxic experiment.

#### 2.2.1. Metal Analysis

For metal element analysis, the sampled filter paper is digested using 4 mL of nitric acid (Merck Ltd. Taiwan, Taipei, Taiwan, 65% GR for Analysis) and 2 mL of hydrofluoric acid (Merck Ltd. Taiwan, Taipei, Taiwan 48% Ultrapur) under microwave digestion at 200 °C for 15 min. The digestive solution was then analyzed using inductively coupled plasma mass spectrometry (ICP-MS) analysis. A total of 34 metals were analyzed including Al, Fe, Na, Mg, K, Ca, Sr, Ba, Ti, Mn, Co, Ni, Cu, Zn, Mo, Cd, Sn, Sb, Tl, Pb, V, Cr, As, Y, Se, Zr, Ge, Rb, Cs, Ga, La, Ce, Nd.

#### 2.2.2. Ion Analysis

The filter paper was sonicated in the ultrasonic oscillator for 90 min. After that, the extracted was filtered through a 0.2 μm pore membrane. The solvent was then ready for further experiment. The ion recovery rates ranged between 88% and 104%, which indicated that this self-developed method has little matrix interference and good recovery. In this study, the water-soluble ion components including Cl^−^, NO_3_^−^, SO_4_^2−^, PO_4_^3−^, Na^+^, NH_4_^+^, K^+^, Ca^2+^, and Mg^2+^ were analyzed using ion chromatograph (IC).

#### 2.2.3. Organic Compound Analysis

One-half of a filter was used for PCDD/F and PAH analysis. After purification by Soxhlet extraction using toluene for PCDD/Fs and hexane for PAHs, the cleaned-up solution was analyzed using a high-resolution gas chromatograph (TRACE GC, Thermo Fisher Scientific Taiwan Co., Ltd., Taiwan) and a high-resolution mass spectrometer (DFS, Thermo Fisher Scientific Taiwan Co., Ltd., Taiwan). The detailed analysis method can be referred to our previous study [24].

### 2.3. Exposure Solution Preparation

The shares of filters used for each extraction are shown in Appendix A. Metal element and organic substance were obtained from a piece of 1/8 of filter paper when the water-soluble ions were extracted from a 1/16 piece of filter paper. Organic components were dissolved in DMSO organic solvent, water-soluble ions were immersed in phosphate buffered saline (PBS). The inorganic metal extraction was dissolved in 50 mL of secondary water (DI water) adjusted to neutral pH before the toxicity test. Blank control samples were extraction of blank filter (non-sampled filter).

### 2.4. Cell Study

Cell model has been proved to be an appropriate model, widely applied in study effect of external factors on human health. Since lung is the first organ to be exposed to atmospheric pollution, we used A549 lung cells to study the toxicity of pollutants. Cells are cultivated in 10 cm diameter petri dish using Ham’s F12 nutrient mixture with 10% fetal bovine serum (FBS) and 1% penicillin/streptomycin antibiotics under 37 °C and 5% CO_2_ condition. Cell subculture is performed every two to three days. Each cell test was performed three times for result confirmation.

#### 2.4.1. Cell Viability

The cell survival rate was evaluated using MTT (3-(4,5-dimethylthiazol-2-yl)-2,5-diphenyltetrazolium bromide) assay. An amount of 100 μL medium including 8 × 10^3^ cells was added into 96-well plate and grown for 24 h. The amount of exposed pollutants equivalent to PM_2.5_ weight (ng) that they were extracted from was shown in Appendix A. Pure DMSO solution was used as control solution. After growing 24 h in exposure solution, cells were incubated with 20 μL of MTT (5 mg/mL-PBS) for 40 min. After being washed by DMSO, the absorbance was quantified using ELISA reader (at wave length of 570–650). The cell viability was calculated using the following equation:$$\%\;viable\;cells=\frac{abs_{samples}-\;abs_{blank}}{abs_{control}-\;abs_{blank}}\;\times \;100$$

#### 2.4.2. Reactive Oxygen Species (ROS) Test

The inflammatory effects caused by oxidative stress were measured in the ROS test. An amount of 5 × 10^5^ cells were exposed to extract solution of 10 μg PM_2.5_ for six hours before being incubated with 2′,7′-dichlorodihydrofluorescein diacetate (DCF-DA) (40 μM) for 30 min. Cells were then washed with PBS and lysed with 200 mM NaOH. Flow cytometry was used to measure the fluorescence intensity.

#### 2.4.3. *Umu* Assay

Genotoxicity effect of PM_2.5_ organic extracts was determined by using *umu* assay. *Umu* assay was performed on *Salmonella typhimurium* strain TA1535 containing NM2009 plasmids. In the first step, bacteria were grown overnight in LB medium containing ampicillin (50 µg/mL). The overnight cultures were then diluted 50 folded with TGA medium (tryptone 1%, glucose 0.2%, NaCl 0.5%) containing ampicillin. The bacteria were cultured until the absorbance at 600 nm reached about 0.21. After that, the β-galactosidase activity expressed by genotoxic activation was measured using 2-nitrophenyl-β-d-galactopyranoside as a target substance, and 2-nitrophenyl-β-d-galactopyranoside as a bacterial growth condition. The expression of the β-galactosidase enzyme was measured by the color of the receptor when it was excited. The light absorbance at wavelength A420 nm was measured. The genotoxicity was examined by the induction ratio (IR) between β-galactosidase activity of sample group and that of control group. IR > 1.5 is considered as presence of genotoxicity effect [25,26].

More detail about the procedure of each test can be found in the Appendix A.

## 3. Results and Discussion

### 3.1. PM_2.5_ Concentrations and Characteristics Measured at Different Stations

The PM_2.5_ concentrations measured from LRT event (Site A), local urban traffic emission (Site B), and night market (Site C) were found to be 39.0 µg/m^3^, 42.9 µg/m^3^, and 28.3 µg/m^3^, respectively (Appendix A). The PM_2.5_ concentrations found in this study (28.3–42.9 µg/m^3^) was lower than that found in central Taiwan (33.2–76.4 μg/m^3^) [27] but higher than a two-year observation of PM_2.5_ concentration at urban area of Taipei (20.5–21.4 µg/m^3^) [20]. We observed highest PM_2.5_ concentrations at traffic area (42.9 µg/m^3^) which was located in the center of Taipei city. Taipei is one of the busiest city in Taiwan with a population of 2.695 million by the end of 2016 [28]. Although Taipei has a quite complete and convenient public transport system, Taipei citizens still depend on private motorcycle for unaccompanied, short-distance, multi-stop trips [29]. However, this PM_2.5_ concentration of Taipei was still lower than that found at other urban areas such as 77–250 μg/m^3^ in Nanjing [30], 62.3 μg/m^3^ in Shanghai [31], 101 μg/m^3^ in Shandong [32].

Not only did possess highest PM_2.5_ concentrations, urban samples also showed highest PCDD/F concentrations of 46.9 fg I-TEQ/m^3^ (Appendix A). This concentration is similar to what was found at Taipei urban area in the previous study where traffic emission contributed the highest proportion of total PCDD/Fs [33]. Despite possessing second highest concentration of PM_2.5_, the total concentrations of PCDD/Fs collected during LRT events was found to be smaller than that of night market samples (7.57 fg I-TEQ/m^3^ comparing to 10.1 fg I-TEQ/m^3^). The PCDD/F concentration during LRT event in this study was found to be much lower than our previous measurement at the same area. PCDD/F concentrations in 2014 and 2015 during LRT event at Fugui Cape station were found to be 16.4–28.9 fg I-TEQ/m^3^ [20].

Low level of total PAHs was observed in LRT (<0.1 BaPeq ng/m^3^) and urban (<1.0 BaPeq ng/m^3^) samples when the concentrations of PAHs during cooking activities were found to be 3.57 BaPeq ng/m^3^. Lower concentrations of PCDD/Fs and PAHs collected from LRT events can be caused by the degradation or deposition of organic compounds during transportation process. High molecular weight carbon can be easily deposited after short transport distance and the can hardly “jump” back to the atmosphere with grasshopper effect [34]. On the other hand, smaller organic compounds can be easily decomposed during the transportation (which take days to reach the receptor areas). In this research, we also found the concentrations of PAHs decrease when the distance to emission sources increased (LRT < traffic < cooking).

The total measured ion concentrations were found to be 10.2 μg/m^3^, 3.44 μg/m^3^, 7.61 μg/m^3^ at LRT, traffic, and coking, respectively (Appendix A). The sum of ion mass accounted for 26.0–28.9% of the total PM_2.5_ mass of LRT and cooking samples, whereas only 8.00% of PM_2.5_ mass from traffic sample was comprised of ions. In PM_2.5_ samples from all three sampling areas, SO_4_^2−^, NH_4_^+^, NO_3_^−^, were the dominant species (Figure 2). Previous research found that these three compounds were major component of particulate matter, especially those with the size <2.5 µm [35]. In PM_2.5_ samples collected from night market, exceptional high proportion of SO_4_^2−^ was recorded (65.5%). This can be explained by meat charbroiling activities since the composition of charcoal can be varied.

Na and K were found to be the main metal elements in LRT and cooking samples (Figure 3, Appendix A). Since Taiwan is an isolated island surrounded by ocean, LRT pollution must travel through the ocean before reaching the island. High concentration of Na and Cl^−^ is the evidence of natural sea salt sources. On the other hand, the use of salt for cooking in the night market also caused elevation of Na concentration in samples collected from this source. In traffic samples, high proportion of aluminum and calcium was the result of crustal dust which was commonly seen on the roads [36]. Iron, as a major exhaust emission from vehicle’s break, also constituted almost 20% of metal element.

### 3.2. Cytotoxicity

#### 3.2.1. Cell Viability

Low cell viability compared to those from blank sample (loss of cell viability = 0) were recorded in metal extractions at all three pollution sources, and organic compound extractions from LRT and cooking samples (Figure 4).

The cell viability can be reduced up to 9% at most after exposing to organic extracts from 0.316 μg PM_2.5_ of LRT and cooking. Metal extracts showed high impact on cell viability even at low amount of PM_2.5_ extract, 1.0 ng (log PM_2.5_~0). Furthermore, in most of the cases, the development of the cells decreased with the increase of concentration of exposed extraction. In previous study, soluble ions and extractable organics compounds were found to decrease the cell viability. The exposed concentration and time showed negative linear relationship with the cell viability [37]. The uniform reduction trend was also observed in most of the samples of our research.

This dose response effect can be an evidence of the causal relationship between exposure to pollution and cell development reduction.

#### 3.2.2. Reactive Oxygen Species (ROS) Test

Figure 5 showed the ROS production rates against different PM_2.5_ extracts from different sources. The exposure dose was equivalent to 10μg PM_2.5._ Compared to extracts of blank samples (ROS production rate = 100%), extracts from PM_2.5_ samples from all three sources showed elevation of ROS in almost all samples. Extracts from different sources showed different effect on ROS induction. Ion and organic extracts from LRT source showed increase of ROS production when organic extracts did not show any elevation of ROS concentration. On the other hand, organic extracts from traffic and cooking sources showed highest ROS induction compared to other types of extract of the same source. Especially, in cooking source, organic extract increase the production rate of ROS up to 150%. This result was similar to previous study when incubating PM_2.5_ with cell culture increased the ROS production [38]. Since ROS is an important factor in increasing cell genotoxicity [38,39], we used organic extract to carry out genotoxicity test.

#### 3.2.3. Genotoxicity Test

For the *umu* assay, we used organic extract as exposure to estimate the genotoxicity of PM_2.5_ extracts. An amount of organic extract equal to that collected from 5 μg and 20 μg of PM_2.5_ were applied separately (Figure 6).

It can be seen that organic compounds from cooking source showed higher β-galactosidase activity than organic compounds from other sources. At low dose (5 μg), the effect difference between different sources were not much differentiated. The induction ratio ranged from 1.1 to 1.3. When the dose increased up to 20 μg, the induction ratio from being exposed to organic extract of cooking activities was observed to reach >1.5, whereas the induction ratio of different exposures did not have much change. The genotoxicity effects of air pollution including PM_2.5_ were reported in the previous studies [38,40]. Previous research using *umu* assay to test genotoxicity showed no significant effect of PAHs collected from sediment [26]. In this study, the effect of our exposure was positive since the exposure compounds were a mixture of different types of organic components.

It can be seen from the above results that organic compounds extracted from cooking samples showed higher impact on both cell viability and ROS concentration. A previous study found that charbroiling yielded more PAHs than other type of cooking [41]. On the other hand, the unregulated combustion process of charcoal could directly emit large amount of pollution. Moreover, bad ventilation in the crowded environment such as the night market could increase the concentration of those pollutions in the atmosphere. Based on Taiwan general public exposure parameter compilation published in 2008, every day, an adult actively inhales an amount of 18.3 m^3^ for male and 12.3 m^3^ for female (Appendix A). The total time of active breathing is estimated to be 15 h (excluding night sleep and nap time). Therefore, every hour, a man breathes in 1.22 m^3^ when a woman breathes in 0.82 m^3^. Assuming that an adult spends two hours per week at the night market under the PM_2.5_ concentrations of 28.3 μg/m^3^ (found in this study), the total exposure to PM_2.5_ from the night market was 69.1 μg in male and 55.4 μg in female. Both of those estimations were higher than the exposure dose of 5–20 μg we applied in cell experiment in this research.

## 4. Conclusions

Even using the same extraction method, the toxicity tests of different sources of pollution were different. This study has verified the protocol for PM_2.5_ toxicity in cell research. It was found that exposure to metal and organic extracts of PM_2.5_ samples from the cooking fumes had the strongest toxic effects on cell viability and cell development. Because of the low exposure concentration, no toxic effects were observed or the toxic effect did not surpass the toxicity threshold. However, the real exposure estimation found that the amount of PM_2.5_ a person can be exposed to can be much higher. This study has confirmed the potential toxicity of fine particulate matter (PM_2.5_) and its component chemicals. Apart from LRT events and traffic activities, cooking should be paid more attention on its effect on human health.

## Figures and Tables

**Figure 1 ijerph-16-05043-f001:**
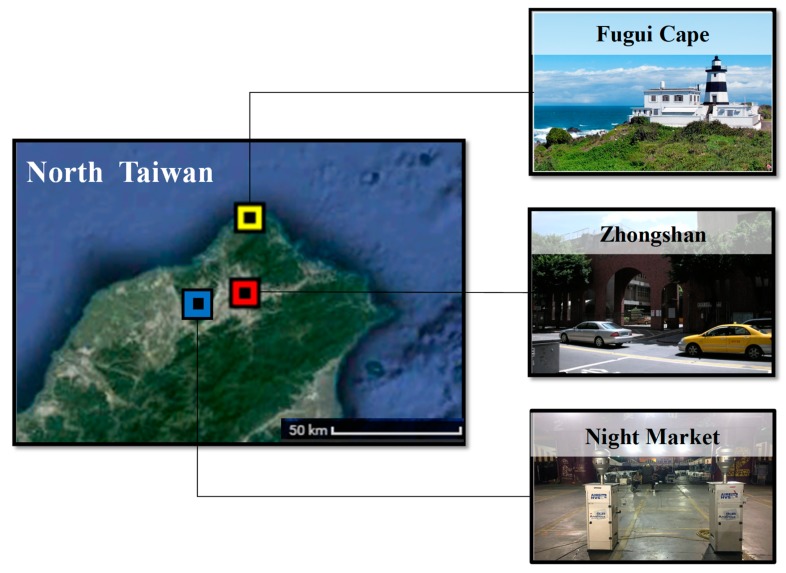
Sampling stations. Fugui Cape—Long-range transport event sampling station, Zhongshan—traffic sampling station, night market—cooking sampling station.

**Figure 2 ijerph-16-05043-f002:**
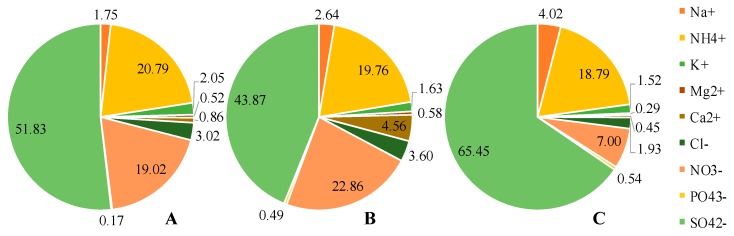
Soluble ion composition (percentage) of (**A**) long-range transport samples, (**B**) traffic samples, (**C**) night market samples.

**Figure 3 ijerph-16-05043-f003:**
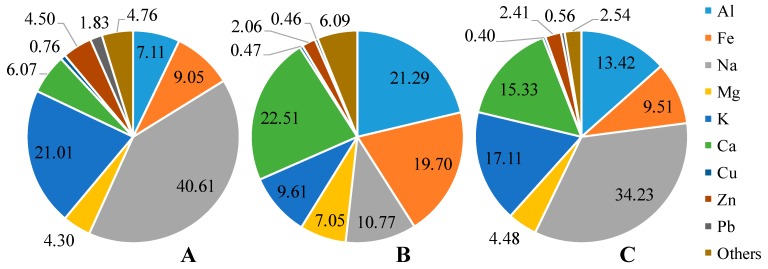
Metal element composition (percentage) of (**A**) long-range transport samples, (**B**) traffic samples, (**C**) night market samples.

**Figure 4 ijerph-16-05043-f004:**
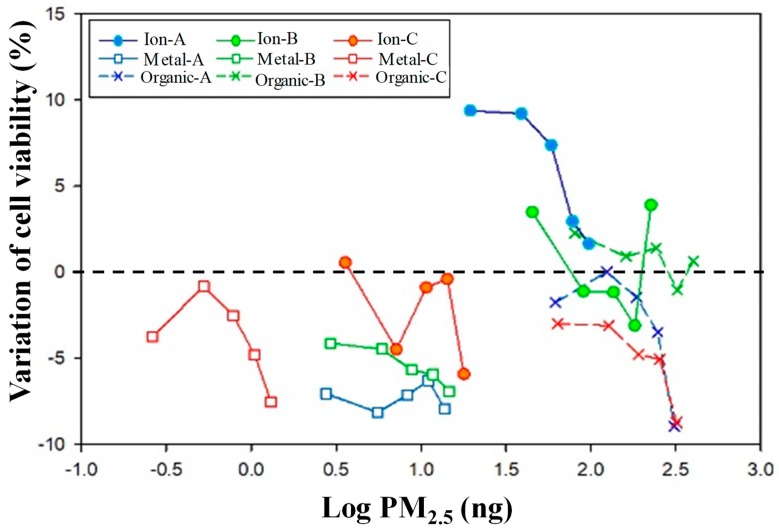
Log PM_2.5_: the equivalent PM_2.5_ concentration corresponding to the concentration of the pollution extract from each sample. A: Long-range transport samples; B: traffic samples; C: night market samples. 0 represents the cell viability when cells were exposed to extraction of negative control sample (extraction of blank filter).

**Figure 5 ijerph-16-05043-f005:**
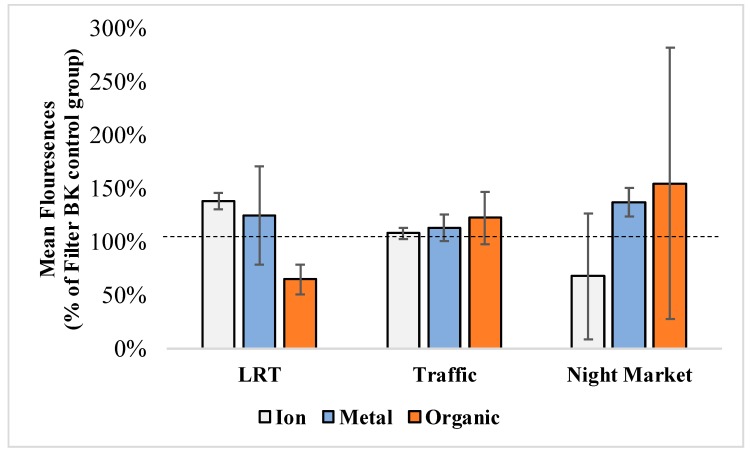
Reactive oxygen species (ROS) expression rate after exposing to 10 μg PM_2.5_ extracts from long-range transport event (LRT), traffic, and cooking in night market sources. The ROS expression rate of blank filter extraction was 100%.

**Figure 6 ijerph-16-05043-f006:**
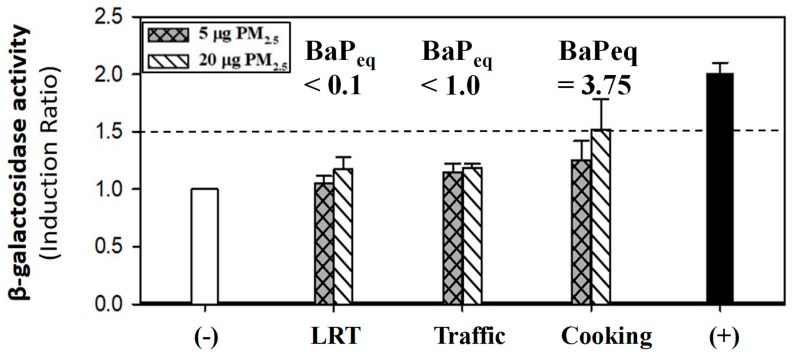
Genotoxicity presented as induction ratio after exposing to PM_2.5_ extracts from LRT, traffic, and night market sources in *umu* assay. Negative control (−) is exposure of blank filter. Positive control (+) is exposure of 10 μM PAHs.

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
