# Peer review of "Cytotoxicity Assessment of PM_2.5_ Collected from Specific Anthropogenic Activities in Taiwan"

_ijerph, 2019, doi:10.3390/ijerph16245043_

Round 1

Reviewer 1 Report

The manuscript is well done. However, I suggest to the authors to improve the findings and results sections because they are weack. In more details, I suggest to compare the findings with previous research and to extend the conclusions.

I suggest also to review the Ennglish language.

Author Response

The manuscript is well done.

However, I suggest to the authors to improve the findings and results sections because they are weak. In more details, I suggest to compare the findings with previous research and to extend the conclusions.

Thank you. We did improve the result and discussion part. Please find it in our revised manuscript

I suggest also to review the English language.

Thank you for your comment. We did review the English in the final manuscript.

Reviewer 2 Report

The authors  studied the cytotoxicity of atmospheric PM2.5 using different tests (MTT assay, Reactive Oxygen Species test and Umu test). The measurements of PM2.5 mass concentration and many of its constituents (metals, ions, organic compounds) have been carried out in three different places representing long range transport, traffic and night-market areas in Taiwan. The issue raised is original and may interest many researchers. However, in my opinion, the findings can provide meaningful reference to further research because current research can be treated as preliminary. It is associated with too few samples taken from each place (two) and analyzed (probably only one but  it is not clearly defined). Small number samples  usually leads to ambiguous  interpretations of results and conclusions.  In addition, the presentation of methodology is not clear in some parts and makes it difficult for readers to understand. The methodological part should be more precise; it is not clear when PM2,5 samples were taken and whether they were representative for the given areas (too few samples). What samplers were used?  The samples were taken for 24 hours, and how to refer it to night samples?

 Please describe the equipment used, origin of test materials, cell culture procedure in more details and all biological tests. Experiment should be  performed in triplicate with three biological replicates. Please give the concentrations of the test samples (not as in l. 151). Similarly, please  define the concentrations for other tests.

Please pay attention on the parts of filters used for analyses. I have currently noticed many inaccuracies (l. 110 and 128, l.134- 135 please check, because half of the filter  was allocated for these analyzes).

There is actually no discussion of the results and current interpretation of the results is hardly justified. Fig. 4 and 5 is illegible. How was this data  calculated?

In conclusion, the study is interesting, however, I have some doubts about the validity of interpretation of the results,  methodological issues,  and the lack of discussion. I would therefore suggest that the paper as it stands should be rejected, but I would encourage resubmission following the  completion of studies.

Author Response

Reviewer 2

The authors studied the cytotoxicity of atmospheric PM2.5 using different tests (MTT assay, Reactive Oxygen Species test and Umu test). The measurements of PM2.5 mass concentration and many of its constituents (metals, ions, organic compounds) have been carried out in three different places representing long range transport, traffic and night-market areas in Taiwan. The issue raised is original and may interest many researchers.

However, in my opinion, the findings can provide meaningful reference to further research because current research can be treated as preliminary. It is associated with too few samples taken from each place (two) and analyzed (probably only one but it is not clearly defined). Small number samples usually lead to ambiguous interpretations of results and conclusions.  

Thank you for your comment. Although we can only collect small number of PM2.5 samples for different events in this study, we tried out best to improve the representativeness of the samples by carefully picked the ordinary day representing the sources of air pollution we want to investigate.

In addition, the presentation of methodology is not clear in some parts and makes it difficult for readers to understand.

Thank you for your comment. The further review of methodology adopted in this study has been completed and added into the revised manuscript for clarification.

The methodological part should be more precise; it is not clear when PM2,5 samples were taken and whether they were representative for the given areas (too few samples).

Thank you for your comment. The PM2.5 samples were collected in 2016. We acknowledge the limitation of small sample size. Therefore, the sampling time was chosen in order to improve the representativeness of our samples. The PM2.5 samples of Long-range transport event were collected on a strong monsoon activity day, traffic samples were collected on normal working days, when night market samples were collected during weekend when around thousand people gathered at the night market for recreation. The detailed can be found in the revised manuscript.

Moreover, since the composition of different components in each station was quite different from each other, we can confirm there is no overlap on the source of pollution.

What samplers were used?  The samples were taken for 24 hours, and how to refer it to night samples?

Thank you for your comment. We made correction in the final manuscript as following:

We used

“In accordance with the European Committee for Standardization standard for PM2.5 (EN 14907), the sampling of PM2.5 was conducted at a flow rate of 500 L min−1. All PM2.5 samples collected at LRT and traffic stations were obtained from 24-hour sampling. Due to the short opening time, the night market samples were only collected for 6 hours. The high-volume samplers for PM2.5 (Analitica AMS® Air Monitoring System, PM2.5-HVS) captured particles on quartz fiber filters (ADVANTEC®, http://www.advantec.tw/, QR-100, ⌀150 mm).”

Please describe the equipment used, origin of test materials, cell culture procedure in more details and all biological tests.

Thank you for your comment. We added the detail of each test in the supplementary materials.

Experiment should be performed in triplicate with three biological replicates.

Yes, we agree and we did perform the experiment three time for each. The standard deviation of each result was added into the manuscript.

Please give the concentrations of the test samples (not as in l. 151). Similarly, please define the concentrations for other tests.

Thank you for your comment. The amount of pm2.5 mass used for each experiment was shown table S2. Due to the difference in PM2.5 collected in three areas within different events, the amount of PM2.5 for each research was somewhat different. We added the following sentence into the revised manuscript.

“The amount of exposed pollutants equivalent to PM2.5 weight (ng) that they were extracted from was shown in Table S2.”

Please pay attention on the parts of filters used for analyses. I have currently noticed many inaccuracies (l. 110 and 128, l.134- 135 please check, because half of the filter was allocated for these analyzes).

Thank you for your comment. We modified and also included Table S1 in the supporting material to avoid misleading.

There is actually no discussion of the results and current interpretation of the results is hardly justified. Fig. 4 and 5 is illegible. How was this data calculated?

We modified Figure 5 and added supporting materials for figure 4 in the Table S4 of Support Material file. We added some relating discussion in the final manuscripts.

In conclusion, the study is interesting, however, I have some doubts about the validity of interpretation of the results, methodological issues, and the lack of discussion. I would therefore suggest that the paper as it stands should be rejected, but I would encourage resubmission following the completion of studies.

Reviewer 3 Report

This article is quite enough descriptive. In my opinion, this article needs minor revision in order to be published. 

First of all, some results needs some references.

In my opinion, a significant absence is the fact that in all results you have the average value without the standard deviation. Normally the results have the form : average value ± standard deviation of values.

In addition you should mention the detection limit of the methods, in different sites ech.

Author Response

Reviewer 3

This article is quite enough descriptive. In my opinion, this article needs minor revision in order to be published. 

First of all, some results need some references.

Thank you for your comment. We did revise this part.

In my opinion, a significant absence is the fact that in all results you have the average value without the standard deviation. Normally the results have the form: average value ± standard deviation of values.

Thank you, we did include the standard deviation.

In addition, you should mention the detection limit of the methods, in different sites each.

Thank you. We added the detection limit into the supplementary materials.

Round 2

Reviewer 2 Report

I accept the corrections made in the article. I'm still not convinced  by the small number of samples analyzed. However, I believe that the quality of the article as a whole has improved.